# The Antimicrobial Effects of Colistin Encapsulated in Chelating Complex Micelles for the Treatment of Multi-Drug-Resistant Gram-Negative Bacteria: A Pharmacokinetic Study

**DOI:** 10.3390/antibiotics12050836

**Published:** 2023-04-30

**Authors:** Wei-Chuan Liao, Chau-Hui Wang, Tzu-Hui Sun, Yu-Cheng Su, Chia-Hung Chen, Wen-Teng Chang, Po-Lin Chen, Yow-Ling Shiue

**Affiliations:** 1Institute of Biomedical Sciences, College of Medicine, National Sun Yat-sen University, Kaohsiung 804201, Taiwan; 2Original Biomedicals Co., Ltd., Tainan 744092, Taiwan; wangch@i-obm.com (C.-H.W.); winnie8012622@gmail.com (T.-H.S.); duncanchen@i-obm.com (C.-H.C.); 3Department of Life Sciences, National Chung Hsing University, Taichung 402202, Taiwan; 4Institute of Basic Medical Sciences, National Cheng Kung University, Tainan 701301, Taiwan; 5Department of Pharmaceutical Science and Technology, Chung Hwa University of Medical Technology, Tainan 717302, Taiwan; 6Department of Internal Medicine, National Cheng Kung University Hospital, College of Medicine, National Cheng Kung University, Tainan 701301, Taiwan; 7Center for Infection Control, National Cheng Kung University Hospital, Tainan 701301, Taiwan; 8Diagnostic Microbiology and Antimicrobial Resistance Laboratory, National Cheng Kung University Hospital, Tainan 701301, Taiwan; 9Department of Microbiology and Immunology, College of Medicine, National Cheng Kung University, Tainan 701301, Taiwan; 10Institute of Precision Medicine, College of Medicine, National Sun Yat-sen University, Kaohsiung 804201, Taiwan

**Keywords:** colistin, chelating complex micelles, multi-drug-resistant Gram-negative bacteria, pharmacokinetics

## Abstract

*Background:* Infections caused by multi-drug-resistant Gram-negative bacteria (MDR-GNB) are an emerging problem globally. Colistin is the last-sort antibiotic for MDR-GNB, but its toxicity limits its clinical use. We aimed to test the efficacy of colistin-loaded micelles (CCM-CL) against drug-resistant *Pseudomonas aeruginosa* and compare their safety with that of free colistin in vitro and in vivo. *Materials and methods:* We incorporated colistin into chelating complex micelles (CCMs), thus producing colistin-loaded micelles (CCM-CL), and conducted both safety and efficacy surveys to elucidate their potential uses. *Results:* In a murine model, the safe dose of CCM-CL was 62.5%, which is much better than that achieved after the intravenous bolus injection of ‘free’ colistin. With a slow drug infusion, the safe dose of CCM-CL reached 16 mg/kg, which is double the free colistin, 8 mg/kg. The area under the curve (AUC) levels for CCM-CL were 4.09- and 4.95-fold higher than those for free colistin in terms of AUC0-t and AUC0-inf, respectively. The elimination half-lives of CCM-CL and free colistin groups were 12.46 and 102.23 min, respectively. In the neutropenic mice model with carbapenem-resistant *Pseudomonas aeruginosa* pneumonia, the 14-day survival rate of the mice treated with CCM-CL was 80%, which was significantly higher than the 30% in the free colistin group (*p* < 0.05). *Conclusions:* Our results showed that CCM-CL, an encapsulated form of colistin, is safe and effective, and thus may become a drug of choice against MDR-GNB.

## 1. Introduction

Multi-drug-resistant (MDR) bacteria are defined as organisms that are resistant to multiple classes of extended-spectrum antimicrobial agents [1]. Infections with multidrug-resistant Gram-negative bacteria (MDR-GNB) are becoming a growing global health concern in all regions of the world as they are associated with increased morbidity, mortality, and healthcare costs [2]. Indeed, the Centers for Disease Control and Prevention have listed several MDR pathogens as ‘urgent threats’. MDR Gram-negative bacteria (GNB) include MDR *Acinetobacter*, extended-spectrum ß-lactamase-producing Enterobacteriaceae, MDR *Pseudomonas aeruginosa*, drug-resistant *Salmonella*, and *Shigella* [3]. It has been estimated that infections associated with carbapenem-resistant *A. baumannii* and Enterobacteriaceae accounted for USD 281 million in U.S. healthcare. In addition, antibiotic choices for MDR-GNB infections pose a severe clinical challenge, as the antibiotic options are limited. Recent efforts to develop antimicrobial agents with novel mechanisms against MDR pathogens have not been able to catch up with the rapidly increasing numbers of infections caused by these pathogens.

Colistin is a lipopeptide antibiotic with activity against many GNB [4,5]. Colistin works by binding to and destabilizing the lipopolysaccharide (LPS) layer of the bacterial outer membrane, resulting in the leakage of cellular contents and ultimately causing bacterial death [6]. It was approved for clinical use in the late 1950s but fell out of favor during the mid-1970s due to concerns regarding its potential nephro- and neurotoxicity [7,8]. Nevertheless, the increasing prevalence of infections caused by MDR-GNB, especially *Pseudomonas aeruginosa*, *Acinetobacter baumannii*, and Klebsiella pneumoniae, led to the reintroduction of colistin in the late 1990s [8,9,10]. However, this did not eradicate the old concerns that had earlier led to the removal of colistin from clinical use. The potential nephrotoxicity and neuronal toxicity of clostin remains the major concern of its clinical use. Higher doses and longer exposure times increase the risk of toxicity. However, lower doses of colistin may affect its therapeutic effect on MDR-GNB infections. Daily doses of less than 9 million IU of colistin are safe for patients with normal renal functions and can achieve high cure rates for infections caused by MDR-GNB [11].

Compared to conventional therapies, the delivery of drugs via a nanosystem improves their efficacy while reducing their potential toxicity. The tiny size of the micelles promotes their extravasation through the endothelium and to the inflammatory site, enabling their effective accumulation in the nidus of the infection [12,13,14]. This nanomedical strategy for improving the delivery of antibiotics may reduce their side effects and the emergence of drug resistance. Chelating complex micelles (CCMs) are composed of drug substances, ferrous ions, and poly(ethylene glycol-b-glutamic acid)s (PEG-*b*-PGA). Drug substances with functional groups that are capable of chelating ferrous ions can be incorporated into CCMs. The carboxylic acid on PGA provides lone pairs of electrons, thus forming coordinate bonds with ferrous ions. The PEG segments without chelating ligands, which spontaneously extend outside the micelles, enhance their dispersal. This approach protects CCMs from non-specific interactions, thus augmenting their survival in the bloodstream. Additionally, the PEG segment allows CCMs to target the infected tissues via an enhanced permeability and retention effect (passive targeting) [15,16,17]. PEG and PGA are both highly biocompatible polymers. In contrast, PEG has been widely used in drug carriers and approved for clinical trials [18,19]. Thus, CCMs have extended the half-life and antiradiation efficacy of amifostine, an old drug, and have received approval from the U.S. Food and Drug Administration for a phase I clinical trial [20,21,22]. This indicates that CCMs reach the safety requirements necessary for research. This study, which focuses on colistin encapsulated in CCMs, was designed to show that this form of the drug can overcome the shortcomings of its free form. Our study represents the first such use of colistin. The comparative assessments of the colistin-loaded CCM (CCM-CL) measured its in vitro antimicrobial effect, safety, pharmacokinetics, and antimicrobial efficacy in a murine model of carbapenem-resistant *P. aeruginosa* pneumonia.

## 2. Results

### 2.1. Characteristics of Colistin-Loaded Micelles

The average size of the CCM-CL was measured with a dynamic light-scattering (DLS) technique; the result indicated that each micelle was of a small uniform dimension of about 27 nm with a PdI of 0.2 (Figure 1). An HPLC also monitored the unloaded colistin. Colistin with a retention time of 5.78 and 9.20 min was quantified, and no free drug remained in the solution. An encapsulation efficiency (EE%) of 100% was therefore calculated from the HPLC using the following formula:Encapsulation efficiency (EE%) = *(Wi − Wf)/Wi* × 100%

*Wi* represents the total quantity of the drug added initially during the preparation. *Wf* is the amount of free drug determined by the HPLC.

### 2.2. Analytical Method Validation for Pharmacokinetics Study

The equations for the calibration curves of colistin A sulfate in rat plasma were y = 2,262,273.079x − 310,048.238; the corresponding equations for colistin B sulfate were y = 2,248,789.626x − 160,682.478. The linearity of the assay was achieved over the range of 0.38–37.91 μg/mL and 0.62–62.09 μg/mL for colistin A and B sulfates, respectively, with coefficients of correlation greater than 0.995.

After back-calculating the concentrations of colistin A and B sulfates in all calibration standards from the derived calibration curve, the accuracy (recovery) results were 90.9% and 122.9%, respectively.

Intra-day precision (RSD%) at concentrations (0.38 μg/mL) of colistin A sulfate ranged between 7.5% and 17.2%, whereas the inter-day precision was 12.6%. Intra-day precision (RSD%) at concentrations (0.62 μg/mL) of colistin B sulfate ranged between 10.7% and 14.0%, whereas the inter-day precision was 12.3%. The results demonstrated acceptable bioanalytical assay accuracy and precision parameters.

### 2.3. MICs of CCM-CL and Free Colistin for P. aeruginosa and A. baumannii

The MIC levels of CCM-CL for the tested strains were identical to those for colistin (Table 1). These results suggested that the CCM-CL steadily releases components against *P. aeruginosa* and *A. baumanii* in vitro.

### 2.4. No Observed Adverse Effect Level (NOAEL)

The bolus doses of free colistin and CCM-CL were 13 and 8 mg/kg, respectively. All the animals survived when free colistin was administered at the dose of 8 mg/kg, but all died at 13 mg/kg. When the drug was administered by slow infusion, the safe dosage of CCM-CL increased to 16 mg/kg, whereas that of the slow injection of free colistin remained at 8 mg/kg (Table 2). As seen in the safety trials, using CCM-CL increased the NOAEL dose by 62.5% with a bolus injection and by 100% with the slow-infusion mode.

### 2.5. Pharmacokinetic Study

The plasma drug concentrations of CCM-CL are shown in Figure 2, and their pharmacokinetic profile is in Table 3. The colistin concentration was approximated in the present study using a one-compartment model, and CCM-CL was calculated using a two-compartment model. The elimination half-time, T1/2(β,) for free colistin was 12.46 min, whereas for CCM-CL it was 102.23 min. The AUC0-t and AUC0-inf for CCM-CL were 4.08- and 4.95-fold higher than those for free colistin, showing the benefit of chelating complex micelles. The elimination half-time and AUC values for CCM-CL are more significant than those for free colistin, indicating that CCM-CL has a higher plasma drug level.

### 2.6. Survival Study in the Murine Pneumonia Model

The 14-day survival rates for the mice treated with CCM-CL, free colistin, and water only were 80%, 30%, and 10%, respectively, as shown in Figure 3. The survival rates of both drug-treated groups were significantly higher than those of the control (water only) group. Moreover, the survival of the group treated with CCM-CL was substantially longer than that of the mice treated with free colistin or water only (*p* < 0.05 and *p* < 0.01, respectively).

Figure 4 shows the colony counts of carbapenem-resistant *P. aeruginosa* (CRPA) isolated from the lungs of sacrificed mice. The colony counts of the mice treated only with water for injection (control group), free colistin, and CCM-CL were 940, 60, and 0 CFU, respectively. These results indicated that using the CCM drug platform enhances the antibacterial effects of free colistin against drug carbapenem-resistant *P. aeruginosa* in a murine model.

## 3. Discussion

In this study, we found that CCM-CL provided a superior safety profile in murine models. The mice could tolerate a dose of CCM-CL that was 62.5% higher than a dose of the free drug. In addition, compared to free colistin, colistin loaded into the new nanosized drug delivery system achieved higher drug concentrations in plasma without increasing toxicity levels. Furthermore, CCM-CL increased the rat survival rate by 50% compared to free colistin in neutropenic mice with CRPA pneumonia. Treatment with CCM-CL benefited mice survival due to the potent killing effect of colistin, which had the best correlation to the AUC/MIC [23]. Our study showed that the AUC level for CCM-CL is about 4–5 times greater than that for free colistin.

Furthermore, the CRPA bacterial load of the infected lungs in the CCM-CL group was far lower than that in the free colistin group, supporting the belief that CCM-CL has a more significant killing effect against CRPA than that of free colistin. Due to the slow release of colistin, CCM-CL can continuously inhibit the growth of *Pseudomonas aeruginosa*, which is partly confirmed by the blood concentration of the drug in this study. This also explains why the survival rate of the CCM-CL group in the animal study was better than that of colistin.

A previous study showed that patients with hyperinflammatory pulmonary disease with acute respiratory distress syndrome had increased transferrin levels [24]. Colistin may be released from CCM-CL once iron is removed by transferrin. Thus, the concentration of colistin presumably reaches a maximum at sites where the transferrin level is high. The colistin concentration is supposed to be elevated in mice treated with CCM-CL for CRPA pneumonia, as iron in the CCMs is chelated in the lungs, where transferrin is highly expressed. However, in the present study, a lower bacterial load of CRPA in the treatment group than that in the colistin group suggested a higher concentration of colistin in the lung. Moreover, a longer elimination half-time makes it possible to administer CCM-CL less frequently than free colistin.

Our study also showed that the overall toxicity of CCM-CL was lower than that of free colistin at higher NOAEL doses in rats. The safety profile indicated that a higher dose achieved a better bactericidal effect. However, the mechanism responsible for the reduced toxicity in rats treated with CCM-CL remains unclear. In the pharmacokinetic study, the serum creatine level and histologic findings of the kidneys in the rats treated with CCM-CL were similar to those observed in the free colistin group (data not shown). These data suggested that CCM-CL does not induce less renal damage than free colistin. Colistin is extensively reabsorbed from the renal tubules and cleared by non-renal mechanisms. Therefore, the reduced toxicity of CCM-CL is likely due to the non-renal mechanism [25].

The major component of the drug carrier, polyglutamic acid (PGA), exists in natural foods as a polymer form of poly-gamma-glutamic acid (γ-PGA), an anionic polypeptide in which a γ-amide linkage polymerizes the α-amino and γ-carboxyl groups of glutamic acid [26]. γ-PGA is naturally synthesized by Bacillus subtilis in fermented soybeans and is present in traditional soy products, such as cheonggukjang in Korea and natto in Japan [27]. An animal study has shown that poly-γ-PGA attenuated inflammasome activation and alleviated the severity of lipopolysaccharide-induced endotoxin shock in mice [28]. Another component of the drug carrier is polyethylene glycol (PEG), and the low-molecular-weight of PEG has been shown to have anti-inflammatory properties. In an animal study, PEG reduced the inflammatory cytokine expression, pyrexia, and mortality by more than 50% in a rat sepsis model [29]. Thus, the low-molecular-weight PEG may play a role in treating severe inflammation and sepsis. Whether PEG-*b*-PGA used as the carrier in the colistin-loaded CCM has similar anti-inflammatory effects in sepsis deserves future research.

In conclusion, CCM-CL, which is a new form of an old drug, has demonstrated both its safety and efficacy in our murine model. In addition, free colistin released from CCM-CL can achieve a more significant concentration-dependent killing effect against MDR-GNB than free colistin delivered by intravenous bolus infusion. CCM-CL would be a safe and effective choice for patients with MDR-GNB infection and needs further validation by clinical research.

## 4. Materials and Methods

### 4.1. Materials

Colistin sulfate (C_53_H_102_N_16_O_17_S, CAS No. 1264-72-8) was purchased from Sigma-Aldrich (St. Louis, MO, USA). Ferrous sulfate heptahydrate (FeSO_4_·7H_2_O) was obtained from SHOWA (Tokyo, Tapan). Purified water was obtained by Milli-Q purification system (Merck, Darmstadt, Germany). Cyclophosphamide, acetonitrile, and trifluoroacetic acid were purchased from Sigma-Aldrich (St. Louis, MO, USA). All other chemicals and reagents were analytical grade and used without further purification.

### 4.2. Preparation of Colistin-Loaded CCM (CCM-CL)

The detailed synthesis procedure of PEG-*b*-PGA and its characterizations were described elsewhere [18]. For the preparation of CCM-CL, colistin sulfate and PEG-*b*-PGA were dissolved in Milli-Q water with stirring. Ferrous sulfate heptahydrate in 0.01 N HCl was then added dropwise to the mixed solution of PEG-*b*-PGA and colistin. The reaction was conducted at room temperature for 2 h and the resulting CCM-CL solution was stored at 2–8 °C until use.

### 4.3. Particle Size Characterization

The size and polydispersity index (PdI) of CCM-CL were evaluated with a dynamic light-scattering technique using Malvern Zetasizer (Nano-ZS, Malvern, UK) at 25 °C. The solution was analyzed in a polystyrene disposable cuvette to obtain the hydrodynamic diameters and PdI of the micelles.

### 4.4. Free Drug Content and Encapsulation Efficiency (EE%)

HPLC measurements were performed on a Thermo Scientific UltiMate 3000 LC system with a Kinetex C18 100Å column (4.6 × 250 mm, 5 micron). A mixture of 0.05% trifluoroacetic acid and methanol (50/50, *v*/*v*) was used as the mobile phase in isocratic elution for 15 min. The flow rate was set to 1 mL/min and the temperature was 25 °C. The sample injection volume of the autosampler was 5.0 μL. The absorbance of the liquid phase at 214 nm was used for the detection of the analytes.

### 4.5. Liquid Chromatography Mass Spectrometry (LC-MS) Conditions for Pharmacokinetics Study

Chromatographic separation was performed by the ACQUITY UHPLC system (Waters Corporation, Milford, MA, USA) on a Phenomenex Kinetex EVO C18 column (150 mm × 4.6 mm; I.D. 5 μm) maintained at 25 ± 5 °C. The mobile phase consisted of solution ‘A’ (acetonitrile) and solution ‘B’ (0.05% trifluoroacetic acid in water, *v*/*v*). The percentage was initially set at 10% and increased to 60% within 5 min, using a flow rate of 0.5 mL/min. After that, the percentage of A quickly decreased to 10% at 5.1 min and remained constant up to 9 min. The total run time was 9 min, and the injection volume was 20 μL. We set the autosampler at 10 ± 5 °C. The ultra-performance liquid chromatography (UHPLC) was interfaced to an ACQUITY QDa detector (Waters Corporation) with electrospray ionization (CDC) in positive mode used for detection. The optimized tuning parameters were as follows: capillary voltage, 0.3 kV; cone voltage, 8 V; and probe temperature, 600 °C. The mass transitions were m/z 585.5 for colistin A, 578.5 for colistin B, and 602.8 for polymyxin B (an internal standard).

### 4.6. Analytical Method Validation for Pharmacokinetics Study

Sample pre-treatment: Five microliters of polymyxin B (Toronto Research Chemicals, CAS number 1405-20-5) dissolved in water (420 μg/mL, an internal standard) was added to 100 μL rat plasma sample. After that, 400 μL of acetonitrile and 10% trichloroacetic acid mixed solution (50/50, *v*/*v*) was added. The samples were vortex-mixed for 1 min and then centrifuged at 12,000 rpm for 10 min. Following centrifugation, the supernatant was collected and filtered with 0.45 μm PVDF membrane filter for analysis.

Linearity: The standard curve was constructed using seven non-zero plasma standards covering the concentrations expected in the study. The original concentrations were then plotted against the responses to obtain the slope, intercept, and correlation coefficient (r^2^) by the least-square linear regression methods.

Accuracy and precision: Accuracy was measured as the percent of deviation from the nominal concentration, whereas precision was determined as the relative standard deviation from the mean (RSD, %). The accuracy results should not be less than 85%. The precision results should not deviate by more than 15% (RSD, % ≤ 15%) except at the lower limit of quantification (LLOQ) where the values should not deviate by more than 20%.

### 4.7. Antimicrobial Effects In Vitro

Antimicrobial effects of the colistin and CCM-CL against *P. aeruginosa* ATCC 27853, *A. baumannii* BCRC 10591, and *P. aeruginosa* 048,431 were determined by the minimal inhibitory concentration (MIC). *P. aeruginosa* ATCC 2785 and *A. baumannii* BCRC 10591 are standard strains; they were purchased from the American Type Culture Collection and Biosource Collection and Research Center, Taiwan, respectively. *P. aeruginosa* 048431 is a carbapenem-resistant blood isolate from a patient with lung adenocarcinoma and ventilator-associated pneumonia. The broth microdilution method determined the MICs for the study drugs. Briefly, serial doubling dilutions of antimicrobial agents were made in a Mueller–Hinton broth. The final bacterial suspension in the mixture was adjusted to 5 × 10^5^ CFUs/mL, and the exact colony count was measured by plating 10-fold serially diluted specimens of 100 μL aliquots on drug-free nutrient agar (Difco Laboratories, Franklin, NJ, USA). The mixtures of bacteria and drugs were incubated at 37 °C for 24 h. The MIC is defined as the lowest antibiotic concentration preventing visible growth in broth [30].

### 4.8. Animals

Sprague–Dawley (SD) rats and C57BL/6 mice were purchased from BioLASCO Taiwan Co., Ltd. (Taipei, Taiwan). The animals were randomized by a stratified sampling method using Excel software (Microsoft^®^). Briefly, the procedures are described as follows: (1) Draw temporary numbers on the tails of 30 mice using a signature pen and enter 1 to 30 in the first line of excel. (2) Enter the formula =R AND () in the second line, and conduct a drop-down copy. (3) Enter the formula = WRAPROWS (SORTBY (A1:A30, B1:B30),3) in the third line of the first cell. (4) Random grouping is finished.

All procedures were approved by the National Cheng Kung University College of Medicine, Chung Hwa University of Medical Technology Animal Care and Use Committee in accordance with the National Institute of Health Guide for the Care and Use of Laboratory Animals and the Animal Welfare Act. The experiments for mice and rats were approved by National Cheng Kung University College of Medicine (IACUC #109307) and Chuang Hua University of Medical Technology (IACUC # A111-03), respectively.

### 4.9. No Observed Adverse Effect Level (NOAEL)

Safety doses of CCM-CL and free colistin in male C57BL/6 mice (21 to 23 g) were determined according to the up-and-down procedure. In the up-and-down method, animals were dosed one at a time. If an animal survived, the dose for the next animal increased; if it died, the next dose decreased. This test determined the safety of a single dose for intravenous bolus injection and 3 min infusion.

### 4.10. Pharmacokinetics Study

The day before the pharmacokinetic study, the right carotid artery of each rat was cannulated for collection of blood samples. Following surgery, rats were individually housed in metabolic cages and were given 24 h for recovery. Six male rats (300 to 320 g) were divided into two groups, and free colistin and CCM-CL were administered intravenously at a dose of 3 mg/kg for 15 min. Blood plasma was collected at min 0, 15, 60, 120, and 180 following drug infusion. All blood samples were collected and transferred into an EDTA-K2 tube, and were then centrifuged at 1500 rcf for 10 min at 4 °C. The experiments were repeated three times. After centrifugation, the plasma was collected and examined by LC-MS. In previous pharmacokinetic studies, colistin is best described by the one-compartment linear model. In contrast, its inactive prodrug, colistin methane sulfonate (CMS), best fits a two-compartment linear model [31,32]. CMS is hydrolyzed to form partially sulfomethylated derivatives as well as colistin sulfate, the active form of the drug [33]. Like CMS, colistin is released from CCM-CL once ferrous ions in the CCM complex are captured by transferrin [34]. Therefore, we estimated the pharmacokinetics for colistin using the one-compartment model and CCM-CL using the two-compartment model. The data were analyzed using PKsolver (Microsoft Excel) [35].

### 4.11. Survival Curve

Male C57BL/6 mice (21 to 23 g) were divided into three groups (namely control, free colistin, and CCM-CL), with 10 mice in each group. All animals were administered 150 mg/kg of cyclophosphamide intraperitoneally, 3 days and 1 day before infection. At day 0 (defined as infection day), mice were infected with 0.04 mL of a drug-resistant strain *P. aeruginosa* 048431, which was administered intratracheally at a concentration of 5 × 10^5^ CFUs/mL. Ninety minutes later, water, free colistin, and CCM-CL were administered intravenously at 8 mg/kg. On day 2, free colistin and CCM-CL at a dose of 8 mg/kg and water were administered again (Figure 5).

### 4.12. Bacterial Load in Lung

This study was conducted and administered along with the survival assay. The pulmonary burden of *P. aeruginosa* was determined on the third day. Mice were sacrificed with carbon dioxide; then, their lungs were removed aseptically and rinsed with sterile saline. The lungs were treated with a homogenizer and then subjected to a 10-fold series of dilutions; 0.05 mL of each dilution was plated on dextrose agar. The plates were then incubated at 37 °C for 2 days, and the colonies were then enumerated.

### 4.13. Statistical Analysis

Data were analyzed using WinSTATt (Microsoft Excel). Overall survival probabilities by age were estimated using the Kaplan–Meier method. The log-rank test (Cox–Mantel) was used to test significant differences in survival among groups.

## Figures and Tables

**Figure 1 antibiotics-12-00836-f001:**
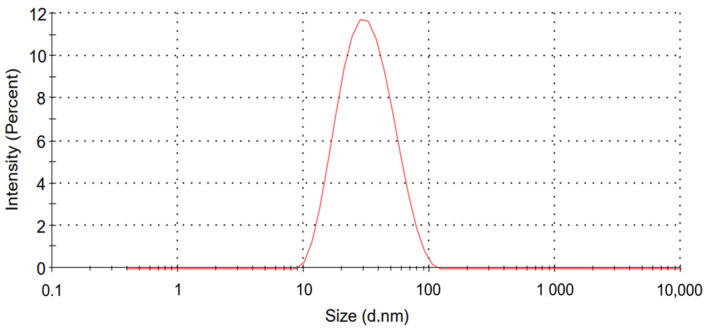
The average size and size distribution of CCM-CL.

**Figure 2 antibiotics-12-00836-f002:**
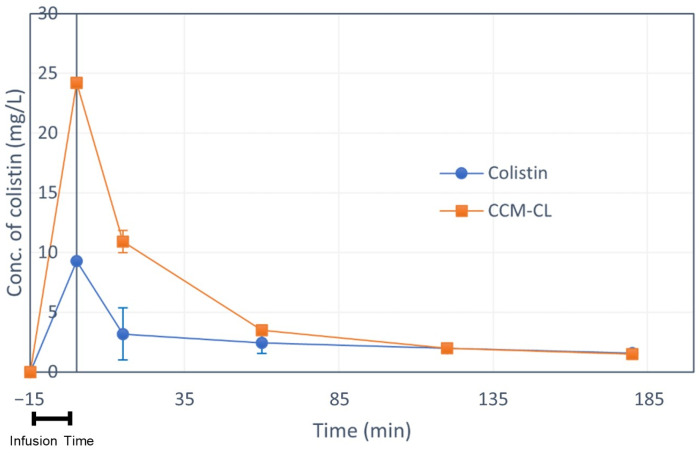
The plasma concentrations of colistin versus time in SD rats treated with intravenous injection of colistin (3 mg/kg) and CCM-CL (3 mg/kg). CCM-CL, orange circles; free colistin, blue circles.

**Figure 3 antibiotics-12-00836-f003:**
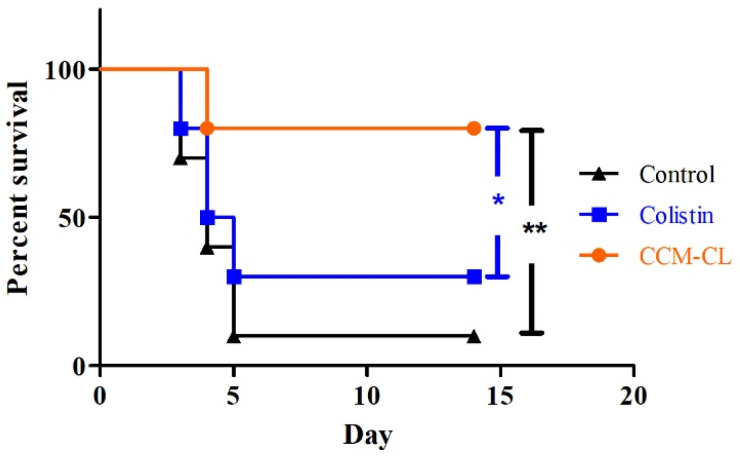
Survival curves of C57BL/6 mice infected intratracheally with *Pseudomonas aeruginosa*. Drugs were administered intravenously 1.5 h after infection (*n* = 10). Black triangles, control; blue squares, colistin; orange circles, CCM-CL. Statistical significance: * *p* < 0.05, ** *p* < 0.01.

**Figure 4 antibiotics-12-00836-f004:**
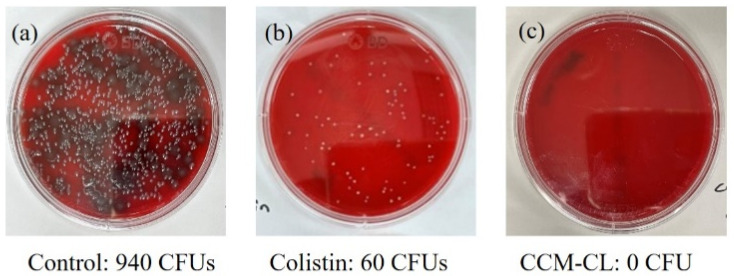
Comparison of antimicrobial effects of CCM-CL and free colistin in mice infected intratracheally with *Pseudomonas aeruginosa*. The mice received drugs intravenously 1.5 h after infection, which were sacrificed 48 h later. The lungs were obtained and homogenated for tissue culture. The bacterial load in the infected lung was measured. Results of 48 h incubation with lung homogenate (**a**–**c**). For culture, lung homogenate was 10-fold diluted.

**Figure 5 antibiotics-12-00836-f005:**
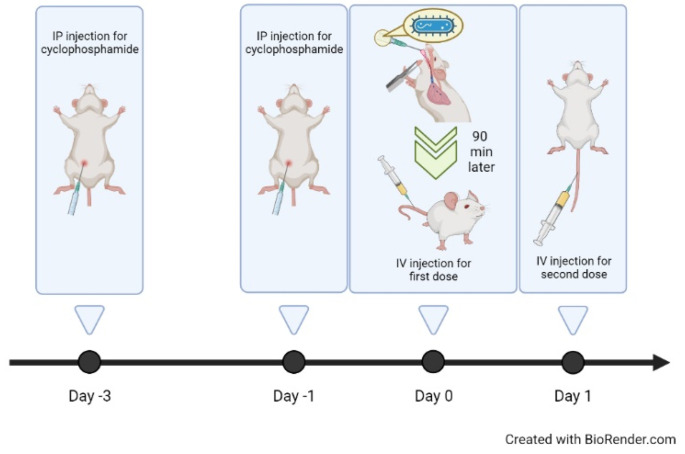
Timeline of survival curve study. Two doses of cyclophosphamide were administered, followed by a pulmonary infection test. Then, water, free colistin, and CCM-CL were administered intravenously 90 min later. The second dose was given the next day.

**Table 1 antibiotics-12-00836-t001:** Minimal inhibitory concentrations of colistin and CCM-CL for *Pseudomonas aeruginosa* and *A. baumannii* isolates.

	Colistin	CCM-CL
*Pseudomonas aeruginosa* ATCC27853	0.5	0.5
*Pseudomonas aeruginosa* 048431	0.5	0.5
*Acinetobacter baumannii* 10591	2	2

**Table 2 antibiotics-12-00836-t002:** No observed adverse effect level dose of colistin and CCM-CL *.

Administration Mode	Samples	Concentration (mg/mL)	Dose (mg/kg)	No. of Survival/Total Mice
Bolus injection	Control	0	0	5/5
	Colistin	6	20	1/3
			13	0/1
			8	5/5
	CCM-CL	6	20	2/3
			13	6/6
Slow infusion	Colistin	6	13	0/1
	CCM-CL	6	20	0/1
			16	2/2

* The test determined the safe single doses for an intravenous bolus injection and that for a 3 min infusion in 8-week-old male C57BL/6 mice.

**Table 3 antibiotics-12-00836-t003:** Values of pharmacokinetic parameters for the intravenous injection of a single dose of colistin and CCM-CL at a dose of 3 mg/kg.

Group	t_1/2_	Tmax	Cmax	AUC_0–t_	AUC_0–inf_
	(min)	(mg/L)	(mg/L)	(mg/L × min)	(mg/L × min)
Colistin	12.46	15	9.03	239.31	239.32
CCM-CL	9.61 (α) 102.23 (β)	15	24.19	976.33	1185.64

AUC, area under the curve; 0–t, time between zero and test; 0–∞, time between zero and infinity; T_1/2_(α), distribution half-time; T_1/2_(β), elimination half-time.

## Data Availability

The data that support the findings of this study are available on request for sresearch or reasonable. business purposes only. The data are not publicly available due to the potential impact on the authors’ commercial interests. Interested parties may contact the Chau-Hui Wang (wangch@i-obm.com) to request access to the data.

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
