# Peer review of "The Antimicrobial Effects of Colistin Encapsulated in Chelating Complex Micelles for the Treatment of Multi-Drug-Resistant Gram-Negative Bacteria: A Pharmacokinetic Study"

_antibiotics, 2023, doi:10.3390/antibiotics12050836_

Round 1

Reviewer 1 Report

The authors presented an interesting study on the biological effects of an encapsulated antibiotic using an animal model. The results are interesting, but there are many questions about the description of the methods.

When developing a method to analyse a bioactive compound in animal blood, at least partial validation of the method must be carried out. This is not present in the manuscript at all.

 In the submitted work on page 4, line 154 reference 22 does not describe a methodology for the quantification of colistin in blood (or plasma). Thus, it is not at all clear how this study was carried out. 

The authors should either provide a reference to a published article, in which the methodology of quantification was developed and validated, or give a full description of the methodology on the equipment used. Without this information, the article cannot be accepted for publication.

The authors write that they used Colistin from Sigma-Aldrich, but there is no CAS number. What exactly was the sample - was it a mixture of antibiotics or an individual compound? The chemical formula of the substance (or mixture of substances) needs to be given.
On page 3, line 119-120, it talks about mass transitions, while m/z values are given for molecular ions, and for double-stranded ones. This fact is neither mentioned nor noted anywhere.

It is not clear what reference 1 on line 118 refers to.

The weight of the animals used in the experiments is not specified; the weight dynamics of the rats in the biological validation experiment are not shown.

In the pharmacokinetics study, it is written that animal blood plasma was used (line 217), whereas the method description (line 154) indicates blood samples. There is no description of how blood was drawn from the animals.

The description of the colistin-containing micelles specifies that the particles are detected from 5.87 to 9.20 min (line 191).  At the same time the samples are only analysed for 9 min. How can this be explained?

Figure 3 shows the antibiotic concentration in animal plasma. The confidence interval is given for two points only. 

And a general remark. In order to confirm the biological effect of colistine-containing micelles, a control experiment had to be carried out on animals in which the biological properties of the micelles not containing the antibiotic were investigated.

Author Response

Reply to reviewer 1

We have answered and revised the manuscript according to the questions proposed by the 5 reviewers. In the reply letter, we listed and answered the questions one by one, and the reviewers could refer the revised MS in which revisions were marked by the underlines. The submitted manuscript was edited and proved by English editing company LetPub®. We would appreciate it very much if you and the reviewers could review the manuscript to be published in your journal.

The authors presented an interesting study on the biological effects of an encapsulated antibiotic using an animal model. The results are interesting, but there are many questions about the description of the methods.

  • When developing a method to analyse a bioactive compound in animal blood, at least partial validation of the method must be carried out. This is not present in the manuscript at all.

Reply: We greatly appreciate your suggestion. We had conducted partial validation for analytical method, and the results will be included in the article. The details are as follows.

Analytical method validation for pharmacokinetic study

Sample pre-treatment Five microliters of polymyxin B (Toronto Research Chemicals, CAS number 1405-20-5) dissolved in water (420 μg/mL, an internal standard) was added to 100 μL rat plasma sample. After that 400 μLof acetonitrile and 10% trichloroacetic acid mixed solution (50/50, v/v) was added. The samples were vortex-mixed for 1 minute and then centrifuged at 12,000 rpm for 10 minutes. Following centrifugation, the supernatant was collected and filtered with 0.45 μm PVDF membrane filter for analysis.

Linearity The standard curve was constructed using seven non-zero plasma standards covering the concentrations expected in the study. The original concentrations were then plotted against the responses to obtain the slope, intercept and correlation coefficient (r2) by the least-square linear regression methods.

Accuracy and precision Accuracy was measured as the percent of deviation from the nominal concentration whereas, the precision was determined as the relative standard deviation from the mean (RSD, %). The accuracy results shouldn’t less than 85%. The precision results shouldn’t deviate by more than 15% (RSD, % ≤ 15%) except at the lower limit of quantification (LLOQ) where the values should not deviate by more than 20%.

Please refer line 144-159.

Validation of drug level was described in the Results. Please refer line 248-261.

The equations for the calibration curves of colistin A sulfate in rat plasma were y = 2262273.079x - 310048.238; the corresponding equations for colistin B sulfate were y = 2248789.626x - 160682.478. The linearity of the assay was achieved over the range of 0.38 – 37.91 μg/mL and 0.62 – 62.09 μg/mL for colistin A and B sulfates, respectively, with coefficients of correlation greater than 0.995.

After back-calculating the concentrations of colistin A and B sulfates in all calibration standards from the derived calibration curve, accuracy (recovery) were 90.9% and 122.9%, respectively.

Intra-day precision (RSD%) at concentrations (0.38 μg/ml) of colistin A sulfate ranged between 7.5% and 17.2% whereas, the inter-day precision was 12.6%. Intra-day precision (RSD%) at concentrations (0.62 μg/mL) of colistin B sulfate ranged between 10.7% and 14.0% whereas, the inter-day precision was 12.3%. The results demonstrated acceptable bioanalytical assay accuracy and precision parameters.

  • In the submitted work on page 4, line 154 reference 22 does not describe a methodology for the quantification of colistin in blood (or plasma). Thus, it is not at all clear how this study was carried out. 

Reply: This reference was incorrectly inserted and deleted in the revised MS.

  • The authors should either provide a reference to a published article, in which the methodology of quantification was developed and validated, or give a full description of the methodology on the equipment used. Without this information, the article cannot be accepted for publication.

Reply: We greatly appreciate your suggestion. The experimental details were addressed in the Materials and Methods. Please refer line 144-159.

  • The authors write that they used Colistin from Sigma-Aldrich, but there is no CAS number. What exactly was the sample - was it a mixture of antibiotics or an individual compound? The chemical formula of the substance (or mixture of substances) needs to be given.

Reply: We greatly appreciate your suggestion. The CAS number of colistin sulfate is 1264-72-8. It is a mixture of compounds, primarily colistin A and B. Its molecular formula is C53H102N16O17S. The information was added. Please refer line 105-106.

  • On page 3, line 119-120, it talks about mass transitions, while m/z values are given for molecular ions, and for double-stranded ones. This fact is neither mentioned nor noted anywhere.

Reply: We greatly appreciate your suggestion. The experimental were described in the Materials and Methods. Please refer line 144-159.

  • It is not clear what reference 1 on line 118 refers to.

Reply: This reference was incorrectly inserted and deleted in the revised MS.

  • The weight of the animals used in the experiments is not specified; the weight dynamics of the rats in the biological validation experiment are not shown.

Reply: Thank you for your suggestion. Relevant animal information was added.

Six male rats (300 to 320 g) were divided into two groups, and free colistin and CCM-CL were administered intravenously at a dose of 3 mg/kg for 15 min. Please refer line 197~199.

Male C57BL/6 mice (21 to 23 g) were divided into three groups (namely control, free colistin, and CCM-CL), 10 mice in each group. Please refer line 212~213.

  • In the pharmacokinetics study, it is written that animal blood plasma was used (line 217), whereas the method description (line 154) indicates blood samples. There is no description of how blood was drawn from the animals.

Reply: Thank you for your suggestion. Description of blood sampling were addressed. Please refer line 195-202.

“The day prior to the pharmacokinetic study, the right carotid artery of each rat was cannulated for collection of blood samples. Following surgery, rats were individually housed in metabolic cages and allowed 24 h for recovery. Six male rats (300 to 320 g) were divided into two groups, and free colistin and CCM-CL were administered intravenously at a dose of 3 mg/kg for 15 min. Blood sample was collected at min 0, 15, 60, 120, and 180 following drug infusion. All blood samples were collected and transferred into an EDTA-K2 tube, and were then centrifuged at 1,500 rcf for 10 min at 4 °C. The experiments were repeated three times.”

  • The description of the colistin-containing micelles specifies that the particles are detected from 5.87 to 9.20 min (line 191).  At the same time the samples are only analysed for 9 min. How can this be explained?

Reply: Thank you for your question. We omitted the description of analytical method for the free drug determination by HPLC in the original MS. The analytical method described below are included in the Materials and Methods. Please refer line 124-129.

“HPLC measurements were performed on a Thermo Scientific UltiMate 3000 LC system with a Kinetex C18 100Å column (4.6 x 250 mm, 5 micron). A mixture of 0.05% trifluoroacetic acid and methanol (50/50, v/v) was used as the mobile phase in isocratic elution for 15 minutes. The flow rate was set to 1 mL/min and the temperature was 25°C. The sample injection volume of the autosampler was 5.0 μL. The absorbance of the liquid phase at 214 nm was used for the detection of the analytes.”

Figure 3 shows the antibiotic concentration in animal plasma. The confidence interval is given for two points only. 

Reply: N=3 in each group. Each dot represents the average value of the 3 repeats.

  • And a general remark. In order to confirm the biological effect of colistine-containing micelles, a control experiment had to be carried out on animals in which the biological properties of the micelles not containing the antibiotic were investigated.

Reply: Thanks for your suggestion. CCM-CLs are formed by the co-complexation of colistin and PEG-b-PGA with ferrous ions. However, PEG-b-PGA and ferrous ions, when used without colistin, are unable to form micelles with the same composition ratio.

Reviewer 2 Report

Dear authors

Thank you for submitting your draft titled "The antimicrobial effects of colistin encapsulated in chelating complex micelles for the treatment of multi-drug-resistant Gram-negative bacteria: a pharmacokinetic study". I appreciate a lot your effort and  I am grateful to submit my comments and suggest changes to improve the paper.

I consider can be published once this is done and by making some minor clarifications in the remarks that I add in the draft in the attached .pdf file.

Congratulations

Author Response

We have answered and revised the manuscript according to the questions proposed by the 5 reviewers. In the reply letter, we listed and answered the questions one by one, and the reviewers could refer the revised MS in which revisions were marked by the underlines. The submitted manuscript was edited and proved by English editing company LetPub®. We would appreciate it very much if you and the reviewers could review the manuscript to be published in your journal.

Reply to reviewer 2

  • Animals
    • Suggest add a table with detailed information animals and experimental groups (descriptive statistics abou somatometry (weigth, sizes, gender)...

Reply: Associated information were added in the Materials and Methods.

“Six male rats (300 to 320 g) were divided into two groups, and free colistin and CCM-CL were administered intravenously at a dose of 3 mg/kg for 15 min.” Please refer line 197~199.

“Male C57BL/6 mice (21 to 23 g) were divided into three groups (namely control, free colistin, and CCM-CL), 10 mice in each group.” Please refer line 212-213.

  • use only males or females rats?

Reply: Male rates were used in our experiment. Please refer line 212-213.

  • Please explain more details about the randomization process...

Reply: Thank you for your question. The randomization procedure is conducted by Excel, and the procedure is described as below. Please refer line 176-181.

  1. Draw temporary numbers on the tails of 30 mice using a signature pen and enter 1 to 30 in the first line of excel.
  2. Next, enter the formula =R AND () in the second line, and conduct a drop-down copy.
  3. in the third line of the first cell enter the formula = WRAPROWS(SORTBY(A1:A30,B1:B30),3)
  4. Random grouping is finished.

  • Pharmacokinetics Study
    • How many replicates were made of the experiments?

Reply: The animal experiments were repeated three times.

  • This happend in your experiments? Recommend determine tranferrin levels in the groups and please show a t test of transferrin leves between treated and control groups.

Reply: Thank you for your question. We did not check the transferrin level in tissues. We proposed that colistin is released from CCM-CL once ferrous ions in the CCM complex are captured by transferrin. A previous study has shown elevation of local transferrin level is induced by severe inflammation by bacterial infection in previous study [ref 30]. We added this assumption in the discussion. Please refer line 338-343.  

  • The data were analyzed using PKsolver (Microsoft Excel). Please add reference Comput Methods Programs Biomed. 2010 Sep;99(3):306-14. doi: 10.1016/j.cmpb.2010.01.007. Epub 2010 Feb 21. PKSolver: An add-in program for pharmacokinetic and pharmacodynamic data analysis in Microsoft Excel. Yong Zhang 1, Meirong Huo, Jianping Zhou, Shaofei Xie Affiliations expand. PMID: 20176408 DOI: 10.1016/j.cmpb.2010.01.007

Reply: Thank you for your suggestion. The reference was added accordingly. Please refer reference 28 (line 210).

  • Result
    • Pharmacokinetic study: Suggest add use Spaghetti plot illustrating the changes... Use as reference to improve this guide: Dunvald AD, Iversen DB, Svendsen ALO, Agergaard K, Kuhlmann IB, Mortensen C, Andersen NE, Järvinen E, Stage TB. Tutorial: Statistical analysis and reporting of clinical pharmacokinetic studies. Clin Transl Sci. 2022 Aug;15(8):1856-1866. doi: 10.1111/cts.13305. Epub 2022 Jun 1. PMID: 35570335; PMCID: PMC9372427.

Reply: Thank you for your suggestion. Figure 3 was revised using Spaghetti plot accordingly.

Discussion

  • Please show a t test of transferrin leves between treated and control groups...

We did not check the transferrin level in tissues. We proposed that colistin is released from CCM-CL once ferrous ions in the CCM complex are captured by transferrin. A previous study has shown elevation of local transferrin level is induced by severe inflammation by bacterial infection in previous study [ref 30]. We this assumption in the discussion. Please refer line 338-343.    

Reviewer 3 Report

The authors tried to examine the antimicrobial effects of colistin encapsulated in chelating 2 complex micelles for the treatment of multi-drug-resistant 3 Gram-negative bacteria as well as its pharmacokinetic profile and to compare those findings with free colistin.

Abstract should have clearly defined aim of the study. It is obvious comparison, but is should be stated.

Introduction: How MDR bacteria is defined?

Aim should be more precisely defined as well as in abstract. Sentences should be more scientific so to speak. It is known what you have done, but it is said complicatedly.

Methods. Have you developed your own method? We need some validation parameters, such as accuracy, precision, LOQ, ets.

Do you have ethical permission to condcut this type of study on animal models?

Pharmacokinetic study. Ref 22 should be checked, doi number could not be connected with this authors. Also, I think that text is not referred to this ref.

Results: explain Cmax and Tmax in intravenous models. Also, how do you explain two-compartment model for CCM-CL

Author Response

We have answered and revised the manuscript according to the questions proposed by the 5 reviewers. In the reply letter, we listed and answered the questions one by one, and the reviewers could refer the revised MS in which revisions were marked by the underlines. The submitted manuscript was edited and proved by English editing company LetPub®. We would appreciate it very much if you and the reviewers could review the manuscript to be published in your journal.

Reply to reviewer 3

  • What is the significance of infection with multidrug-resistant Gram-negative bacteria (MDR-GNB) on a global scale?

Reply: Infection with multidrug-resistant Gram-negative bacteria (MDR-GNB) is a growing global health concern in all regions of the world as it is associated with increased morbidity, mortality, and healthcare costs. Please refer line 51~55.

  • What is the mechanism of action of colistin against MDR-GNB?

Reply: Colistin works by binding to and destabilize the lipopolysaccharide (LPS) layer of the bacterial outer membrane, resulting in leakage of cellular contents and ultimately causing bacterial death. Please refer line 64~67.

  • To what extent does the toxicity of colistin limit its clinical use?

Reply: The potential nephrotoxicity and neuronal toxicity of colistin remains the major concern of its clinical use. Higher dose and longer exposure times increase the risk of toxicity. However, colistin of lower dose may affect its therapeutic effect on MDR-GNB infections. Daily dose of less than 9 million IU of colistin is safe in patients with normal renal function and can achieve high cure rate for infections caused by MDR-GNB. Please refer line 72~77.

  • What are chelating complex micelles (CCMs), and how can they be used in the administration of colistin?

Reply: Thank you for your question. Chelating complex micelles (CCMs) are composed of drug substances, ferrous ions, and poly(ethylene glycol-b-glutamic acid)s (PEG-b-PGA). Drug substances with functional groups capable of chelating ferrous ions can be incorporated into CCMs. Please refer line 83-86.

  • What was the safe dose of CCM-CL in the mouse model, and how did it compare to the safe dose achieved with free colistin?

Reply: Thank you for your question. This is addressed in the No-observed-adverse-effect level (NOAEL) section of Result. The safe doses of Colistin and CCM-CL are 8 mg/kg and 13 mg/kg, respectively, when administered by bolus. As seen in the safety trials, using CCM-CL increased the NOAEL dose by 62.5% with bolus injection and by 100% with the slow-infusion mode. Please refer line 269-274.

  • What was the elimination half-life of CCM-CL compared to free colistin in the study?

Reply: Thank you for your question. This is stated in the Pharmacokinetic study section of Result. The half-lives of Colistin and CCM-CL are 12.46 and 102.23 minutes, respectively. Please refer line 282-283.

  • What was the 14-day survival rate of mice treated with CCM-CL compared with that of mice treated with free colistin in the neutropenic mouse model of carbapenem-resistant Pseudomonas aeruginosa pneumonia?

Reply: This is written in the Survival study in the murine pneumonia model section of Result. The 14-day survival rates for the mice treated with CCM-CL, free colistin, and water only were 80%, 30%, and 10%, respectively. Please refer line 297-298.

  • Based on the results of the study, how might CCM-CL become a drug of choice for MDR-GNB?

Reply: Thank you for your question. We conclude that CCM-CL would be a safe and effective choice for patients with MDR-GNB infection based on the good safety profile and its effectiveness against MDR-GNB infection. Please refer line 369-374.

What are the potential implications of the study results for the treatment of MDR-GNB infections in humans?

Reply: We conclude that CCM-CL would be a safe and effective choice for patients with MDR-GNB infection. Further clinical trials are warranted. Please refer line 372-374.

  • What further research might be needed to confirm the safety and effectiveness of CCM-CL in humans?

Reply: Thank you for your question. CCM-CL needs to be confirmed its safety and effectiveness for patients with MDR-GNB sepsis in future clinical trials. Please refer line 372-374.

  • How small and uniform were the dimensions of the individual micelles after the DLS technique?

Reply: The results of DLS measurements indicated that CCM-CL had an average size of 27 nm with a narrow size distribution, as evidenced by a polydispersity index (PdI) of 0.2. Please refer line 237-238.

  • What is the formula for calculating the encapsulation efficiency (EE %)?

Reply: As shown on page 6, line 240-244, an encapsulation efficiency (EE%) was calculated using the following formula:

Encapsulation efficiency (EE%) = (Wi-Wf)/Wi × 100%

Wi represents the total quantity of drug added initially during preparation. Wf is the amount of free drug determined by HPLC.

  • How does CCM-CL compare to free colistin in terms of bacterial load and killing efficacy against CRPA?

Reply: Thank you for your question. The microbiological results were shown in Figure 5. Pulmonary bacterial burden of CCM-CL group was 0 CFU and colistin was 60 CFU at the same dose. Please refer line 309-311.

  • Why is a lower bacterial load of CRPA significant in the treatment group compared to the colistin group?

Reply: Due to slow release of colistin, CCM-CL can continuously inhibit the growth of Pseudomonas aeruginosa, which is partly confirmed by the blood concentration of the drug in this study. This also explains why the survival rate of the CCM-CL group in the animal study was better than that of Colistin. We added this explanation in the Discussion. Please refer line 332-336.

Reply to reviewer 3

  • What is the significance of infection with multidrug-resistant Gram-negative bacteria (MDR-GNB) on a global scale?

Reply: Infection with multidrug-resistant Gram-negative bacteria (MDR-GNB) is a growing global health concern in all regions of the world as it is associated with increased morbidity, mortality, and healthcare costs. Please refer line 51~55.

  • What is the mechanism of action of colistin against MDR-GNB?

Reply: Colistin works by binding to and destabilize the lipopolysaccharide (LPS) layer of the bacterial outer membrane, resulting in leakage of cellular contents and ultimately causing bacterial death. Please refer line 64~67.

  • To what extent does the toxicity of colistin limit its clinical use?

Reply: The potential nephrotoxicity and neuronal toxicity of colistin remains the major concern of its clinical use. Higher dose and longer exposure times increase the risk of toxicity. However, colistin of lower dose may affect its therapeutic effect on MDR-GNB infections. Daily dose of less than 9 million IU of colistin is safe in patients with normal renal function and can achieve high cure rate for infections caused by MDR-GNB. Please refer line 72~77.

  • What are chelating complex micelles (CCMs), and how can they be used in the administration of colistin?

Reply: Thank you for your question. Chelating complex micelles (CCMs) are composed of drug substances, ferrous ions, and poly(ethylene glycol-b-glutamic acid)s (PEG-b-PGA). Drug substances with functional groups capable of chelating ferrous ions can be incorporated into CCMs. Please refer line 83-86.

  • What was the safe dose of CCM-CL in the mouse model, and how did it compare to the safe dose achieved with free colistin?

Reply: Thank you for your question. This is addressed in the No-observed-adverse-effect level (NOAEL) section of Result. The safe doses of Colistin and CCM-CL are 8 mg/kg and 13 mg/kg, respectively, when administered by bolus. As seen in the safety trials, using CCM-CL increased the NOAEL dose by 62.5% with bolus injection and by 100% with the slow-infusion mode. Please refer line 269-274.

  • What was the elimination half-life of CCM-CL compared to free colistin in the study?

Reply: Thank you for your question. This is stated in the Pharmacokinetic study section of Result. The half-lives of Colistin and CCM-CL are 12.46 and 102.23 minutes, respectively. Please refer line 282-283.

  • What was the 14-day survival rate of mice treated with CCM-CL compared with that of mice treated with free colistin in the neutropenic mouse model of carbapenem-resistant Pseudomonas aeruginosa pneumonia?

Reply: This is written in the Survival study in the murine pneumonia model section of Result. The 14-day survival rates for the mice treated with CCM-CL, free colistin, and water only were 80%, 30%, and 10%, respectively. Please refer line 297-298.

  • Based on the results of the study, how might CCM-CL become a drug of choice for MDR-GNB?

Reply: Thank you for your question. We conclude that CCM-CL would be a safe and effective choice for patients with MDR-GNB infection based on the good safety profile and its effectiveness against MDR-GNB infection. Please refer line 369-374.

What are the potential implications of the study results for the treatment of MDR-GNB infections in humans?

Reply: We conclude that CCM-CL would be a safe and effective choice for patients with MDR-GNB infection. Further clinical trials are warranted. Please refer line 372-374.

  • What further research might be needed to confirm the safety and effectiveness of CCM-CL in humans?

Reply: Thank you for your question. CCM-CL needs to be confirmed its safety and effectiveness for patients with MDR-GNB sepsis in future clinical trials. Please refer line 372-374.

  • How small and uniform were the dimensions of the individual micelles after the DLS technique?

Reply: The results of DLS measurements indicated that CCM-CL had an average size of 27 nm with a narrow size distribution, as evidenced by a polydispersity index (PdI) of 0.2. Please refer line 237-238.

  • What is the formula for calculating the encapsulation efficiency (EE %)?

Reply: As shown on page 6, line 240-244, an encapsulation efficiency (EE%) was calculated using the following formula:

Encapsulation efficiency (EE%) = (Wi-Wf)/Wi × 100%

Wi represents the total quantity of drug added initially during preparation. Wf is the amount of free drug determined by HPLC.

  • How does CCM-CL compare to free colistin in terms of bacterial load and killing efficacy against CRPA?

Reply: Thank you for your question. The microbiological results were shown in Figure 5. Pulmonary bacterial burden of CCM-CL group was 0 CFU and colistin was 60 CFU at the same dose. Please refer line 309-311.

  • Why is a lower bacterial load of CRPA significant in the treatment group compared to the colistin group?

Reply: Due to slow release of colistin, CCM-CL can continuously inhibit the growth of Pseudomonas aeruginosa, which is partly confirmed by the blood concentration of the drug in this study. This also explains why the survival rate of the CCM-CL group in the animal study was better than that of Colistin. We added this explanation in the Discussion. Please refer line 332-336.

Reply to reviewer 3

  • What is the significance of infection with multidrug-resistant Gram-negative bacteria (MDR-GNB) on a global scale?

Reply: Infection with multidrug-resistant Gram-negative bacteria (MDR-GNB) is a growing global health concern in all regions of the world as it is associated with increased morbidity, mortality, and healthcare costs. Please refer line 51~55.

  • What is the mechanism of action of colistin against MDR-GNB?

Reply: Colistin works by binding to and destabilize the lipopolysaccharide (LPS) layer of the bacterial outer membrane, resulting in leakage of cellular contents and ultimately causing bacterial death. Please refer line 64~67.

  • To what extent does the toxicity of colistin limit its clinical use?

Reply: The potential nephrotoxicity and neuronal toxicity of colistin remains the major concern of its clinical use. Higher dose and longer exposure times increase the risk of toxicity. However, colistin of lower dose may affect its therapeutic effect on MDR-GNB infections. Daily dose of less than 9 million IU of colistin is safe in patients with normal renal function and can achieve high cure rate for infections caused by MDR-GNB. Please refer line 72~77.

  • What are chelating complex micelles (CCMs), and how can they be used in the administration of colistin?

Reply: Thank you for your question. Chelating complex micelles (CCMs) are composed of drug substances, ferrous ions, and poly(ethylene glycol-b-glutamic acid)s (PEG-b-PGA). Drug substances with functional groups capable of chelating ferrous ions can be incorporated into CCMs. Please refer line 83-86.

  • What was the safe dose of CCM-CL in the mouse model, and how did it compare to the safe dose achieved with free colistin?

Reply: Thank you for your question. This is addressed in the No-observed-adverse-effect level (NOAEL) section of Result. The safe doses of Colistin and CCM-CL are 8 mg/kg and 13 mg/kg, respectively, when administered by bolus. As seen in the safety trials, using CCM-CL increased the NOAEL dose by 62.5% with bolus injection and by 100% with the slow-infusion mode. Please refer line 269-274.

  • What was the elimination half-life of CCM-CL compared to free colistin in the study?

Reply: Thank you for your question. This is stated in the Pharmacokinetic study section of Result. The half-lives of Colistin and CCM-CL are 12.46 and 102.23 minutes, respectively. Please refer line 282-283.

  • What was the 14-day survival rate of mice treated with CCM-CL compared with that of mice treated with free colistin in the neutropenic mouse model of carbapenem-resistant Pseudomonas aeruginosa pneumonia?

Reply: This is written in the Survival study in the murine pneumonia model section of Result. The 14-day survival rates for the mice treated with CCM-CL, free colistin, and water only were 80%, 30%, and 10%, respectively. Please refer line 297-298.

  • Based on the results of the study, how might CCM-CL become a drug of choice for MDR-GNB?

Reply: Thank you for your question. We conclude that CCM-CL would be a safe and effective choice for patients with MDR-GNB infection based on the good safety profile and its effectiveness against MDR-GNB infection. Please refer line 369-374.

What are the potential implications of the study results for the treatment of MDR-GNB infections in humans?

Reply: We conclude that CCM-CL would be a safe and effective choice for patients with MDR-GNB infection. Further clinical trials are warranted. Please refer line 372-374.

  • What further research might be needed to confirm the safety and effectiveness of CCM-CL in humans?

Reply: Thank you for your question. CCM-CL needs to be confirmed its safety and effectiveness for patients with MDR-GNB sepsis in future clinical trials. Please refer line 372-374.

  • How small and uniform were the dimensions of the individual micelles after the DLS technique?

Reply: The results of DLS measurements indicated that CCM-CL had an average size of 27 nm with a narrow size distribution, as evidenced by a polydispersity index (PdI) of 0.2. Please refer line 237-238.

  • What is the formula for calculating the encapsulation efficiency (EE %)?

Reply: As shown on page 6, line 240-244, an encapsulation efficiency (EE%) was calculated using the following formula:

Encapsulation efficiency (EE%) = (Wi-Wf)/Wi × 100%

Wi represents the total quantity of drug added initially during preparation. Wf is the amount of free drug determined by HPLC.

  • How does CCM-CL compare to free colistin in terms of bacterial load and killing efficacy against CRPA?

Reply: Thank you for your question. The microbiological results were shown in Figure 5. Pulmonary bacterial burden of CCM-CL group was 0 CFU and colistin was 60 CFU at the same dose. Please refer line 309-311.

  • Why is a lower bacterial load of CRPA significant in the treatment group compared to the colistin group?

Reply: Due to slow release of colistin, CCM-CL can continuously inhibit the growth of Pseudomonas aeruginosa, which is partly confirmed by the blood concentration of the drug in this study. This also explains why the survival rate of the CCM-CL group in the animal study was better than that of Colistin. We added this explanation in the Discussion. Please refer line 332-336.

Reply to reviewer 3

  • What is the significance of infection with multidrug-resistant Gram-negative bacteria (MDR-GNB) on a global scale?

Reply: Infection with multidrug-resistant Gram-negative bacteria (MDR-GNB) is a growing global health concern in all regions of the world as it is associated with increased morbidity, mortality, and healthcare costs. Please refer line 51~55.

  • What is the mechanism of action of colistin against MDR-GNB?

Reply: Colistin works by binding to and destabilize the lipopolysaccharide (LPS) layer of the bacterial outer membrane, resulting in leakage of cellular contents and ultimately causing bacterial death. Please refer line 64~67.

  • To what extent does the toxicity of colistin limit its clinical use?

Reply: The potential nephrotoxicity and neuronal toxicity of colistin remains the major concern of its clinical use. Higher dose and longer exposure times increase the risk of toxicity. However, colistin of lower dose may affect its therapeutic effect on MDR-GNB infections. Daily dose of less than 9 million IU of colistin is safe in patients with normal renal function and can achieve high cure rate for infections caused by MDR-GNB. Please refer line 72~77.

  • What are chelating complex micelles (CCMs), and how can they be used in the administration of colistin?

Reply: Thank you for your question. Chelating complex micelles (CCMs) are composed of drug substances, ferrous ions, and poly(ethylene glycol-b-glutamic acid)s (PEG-b-PGA). Drug substances with functional groups capable of chelating ferrous ions can be incorporated into CCMs. Please refer line 83-86.

  • What was the safe dose of CCM-CL in the mouse model, and how did it compare to the safe dose achieved with free colistin?

Reply: Thank you for your question. This is addressed in the No-observed-adverse-effect level (NOAEL) section of Result. The safe doses of Colistin and CCM-CL are 8 mg/kg and 13 mg/kg, respectively, when administered by bolus. As seen in the safety trials, using CCM-CL increased the NOAEL dose by 62.5% with bolus injection and by 100% with the slow-infusion mode. Please refer line 269-274.

  • What was the elimination half-life of CCM-CL compared to free colistin in the study?

Reply: Thank you for your question. This is stated in the Pharmacokinetic study section of Result. The half-lives of Colistin and CCM-CL are 12.46 and 102.23 minutes, respectively. Please refer line 282-283.

  • What was the 14-day survival rate of mice treated with CCM-CL compared with that of mice treated with free colistin in the neutropenic mouse model of carbapenem-resistant Pseudomonas aeruginosa pneumonia?

Reply: This is written in the Survival study in the murine pneumonia model section of Result. The 14-day survival rates for the mice treated with CCM-CL, free colistin, and water only were 80%, 30%, and 10%, respectively. Please refer line 297-298.

  • Based on the results of the study, how might CCM-CL become a drug of choice for MDR-GNB?

Reply: Thank you for your question. We conclude that CCM-CL would be a safe and effective choice for patients with MDR-GNB infection based on the good safety profile and its effectiveness against MDR-GNB infection. Please refer line 369-374.

What are the potential implications of the study results for the treatment of MDR-GNB infections in humans?

Reply: We conclude that CCM-CL would be a safe and effective choice for patients with MDR-GNB infection. Further clinical trials are warranted. Please refer line 372-374.

  • What further research might be needed to confirm the safety and effectiveness of CCM-CL in humans?

Reply: Thank you for your question. CCM-CL needs to be confirmed its safety and effectiveness for patients with MDR-GNB sepsis in future clinical trials. Please refer line 372-374.

  • How small and uniform were the dimensions of the individual micelles after the DLS technique?

Reply: The results of DLS measurements indicated that CCM-CL had an average size of 27 nm with a narrow size distribution, as evidenced by a polydispersity index (PdI) of 0.2. Please refer line 237-238.

  • What is the formula for calculating the encapsulation efficiency (EE %)?

Reply: As shown on page 6, line 240-244, an encapsulation efficiency (EE%) was calculated using the following formula:

Encapsulation efficiency (EE%) = (Wi-Wf)/Wi × 100%

Wi represents the total quantity of drug added initially during preparation. Wf is the amount of free drug determined by HPLC.

  • How does CCM-CL compare to free colistin in terms of bacterial load and killing efficacy against CRPA?

Reply: Thank you for your question. The microbiological results were shown in Figure 5. Pulmonary bacterial burden of CCM-CL group was 0 CFU and colistin was 60 CFU at the same dose. Please refer line 309-311.

  • Why is a lower bacterial load of CRPA significant in the treatment group compared to the colistin group?

Reply: Due to slow release of colistin, CCM-CL can continuously inhibit the growth of Pseudomonas aeruginosa, which is partly confirmed by the blood concentration of the drug in this study. This also explains why the survival rate of the CCM-CL group in the animal study was better than that of Colistin. We added this explanation in the Discussion. Please refer line 332-336.

Reply to reviewer 3

  • What is the significance of infection with multidrug-resistant Gram-negative bacteria (MDR-GNB) on a global scale?

Reply: Infection with multidrug-resistant Gram-negative bacteria (MDR-GNB) is a growing global health concern in all regions of the world as it is associated with increased morbidity, mortality, and healthcare costs. Please refer line 51~55.

  • What is the mechanism of action of colistin against MDR-GNB?

Reply: Colistin works by binding to and destabilize the lipopolysaccharide (LPS) layer of the bacterial outer membrane, resulting in leakage of cellular contents and ultimately causing bacterial death. Please refer line 64~67.

  • To what extent does the toxicity of colistin limit its clinical use?

Reply: The potential nephrotoxicity and neuronal toxicity of colistin remains the major concern of its clinical use. Higher dose and longer exposure times increase the risk of toxicity. However, colistin of lower dose may affect its therapeutic effect on MDR-GNB infections. Daily dose of less than 9 million IU of colistin is safe in patients with normal renal function and can achieve high cure rate for infections caused by MDR-GNB. Please refer line 72~77.

  • What are chelating complex micelles (CCMs), and how can they be used in the administration of colistin?

Reply: Thank you for your question. Chelating complex micelles (CCMs) are composed of drug substances, ferrous ions, and poly(ethylene glycol-b-glutamic acid)s (PEG-b-PGA). Drug substances with functional groups capable of chelating ferrous ions can be incorporated into CCMs. Please refer line 83-86.

  • What was the safe dose of CCM-CL in the mouse model, and how did it compare to the safe dose achieved with free colistin?

Reply: Thank you for your question. This is addressed in the No-observed-adverse-effect level (NOAEL) section of Result. The safe doses of Colistin and CCM-CL are 8 mg/kg and 13 mg/kg, respectively, when administered by bolus. As seen in the safety trials, using CCM-CL increased the NOAEL dose by 62.5% with bolus injection and by 100% with the slow-infusion mode. Please refer line 269-274.

  • What was the elimination half-life of CCM-CL compared to free colistin in the study?

Reply: Thank you for your question. This is stated in the Pharmacokinetic study section of Result. The half-lives of Colistin and CCM-CL are 12.46 and 102.23 minutes, respectively. Please refer line 282-283.

  • What was the 14-day survival rate of mice treated with CCM-CL compared with that of mice treated with free colistin in the neutropenic mouse model of carbapenem-resistant Pseudomonas aeruginosa pneumonia?

Reply: This is written in the Survival study in the murine pneumonia model section of Result. The 14-day survival rates for the mice treated with CCM-CL, free colistin, and water only were 80%, 30%, and 10%, respectively. Please refer line 297-298.

  • Based on the results of the study, how might CCM-CL become a drug of choice for MDR-GNB?

Reply: Thank you for your question. We conclude that CCM-CL would be a safe and effective choice for patients with MDR-GNB infection based on the good safety profile and its effectiveness against MDR-GNB infection. Please refer line 369-374.

What are the potential implications of the study results for the treatment of MDR-GNB infections in humans?

Reply: We conclude that CCM-CL would be a safe and effective choice for patients with MDR-GNB infection. Further clinical trials are warranted. Please refer line 372-374.

  • What further research might be needed to confirm the safety and effectiveness of CCM-CL in humans?

Reply: Thank you for your question. CCM-CL needs to be confirmed its safety and effectiveness for patients with MDR-GNB sepsis in future clinical trials. Please refer line 372-374.

  • How small and uniform were the dimensions of the individual micelles after the DLS technique?

Reply: The results of DLS measurements indicated that CCM-CL had an average size of 27 nm with a narrow size distribution, as evidenced by a polydispersity index (PdI) of 0.2. Please refer line 237-238.

  • What is the formula for calculating the encapsulation efficiency (EE %)?

Reply: As shown on page 6, line 240-244, an encapsulation efficiency (EE%) was calculated using the following formula:

Encapsulation efficiency (EE%) = (Wi-Wf)/Wi × 100%

Wi represents the total quantity of drug added initially during preparation. Wf is the amount of free drug determined by HPLC.

  • How does CCM-CL compare to free colistin in terms of bacterial load and killing efficacy against CRPA?

Reply: Thank you for your question. The microbiological results were shown in Figure 5. Pulmonary bacterial burden of CCM-CL group was 0 CFU and colistin was 60 CFU at the same dose. Please refer line 309-311.

  • Why is a lower bacterial load of CRPA significant in the treatment group compared to the colistin group?

Reply: Due to slow release of colistin, CCM-CL can continuously inhibit the growth of Pseudomonas aeruginosa, which is partly confirmed by the blood concentration of the drug in this study. This also explains why the survival rate of the CCM-CL group in the animal study was better than that of Colistin. We added this explanation in the Discussion. Please refer line 332-336.

Reply to reviewer 3

  • What is the significance of infection with multidrug-resistant Gram-negative bacteria (MDR-GNB) on a global scale?

Reply: Infection with multidrug-resistant Gram-negative bacteria (MDR-GNB) is a growing global health concern in all regions of the world as it is associated with increased morbidity, mortality, and healthcare costs. Please refer line 51~55.

  • What is the mechanism of action of colistin against MDR-GNB?

Reply: Colistin works by binding to and destabilize the lipopolysaccharide (LPS) layer of the bacterial outer membrane, resulting in leakage of cellular contents and ultimately causing bacterial death. Please refer line 64~67.

  • To what extent does the toxicity of colistin limit its clinical use?

Reply: The potential nephrotoxicity and neuronal toxicity of colistin remains the major concern of its clinical use. Higher dose and longer exposure times increase the risk of toxicity. However, colistin of lower dose may affect its therapeutic effect on MDR-GNB infections. Daily dose of less than 9 million IU of colistin is safe in patients with normal renal function and can achieve high cure rate for infections caused by MDR-GNB. Please refer line 72~77.

  • What are chelating complex micelles (CCMs), and how can they be used in the administration of colistin?

Reply: Thank you for your question. Chelating complex micelles (CCMs) are composed of drug substances, ferrous ions, and poly(ethylene glycol-b-glutamic acid)s (PEG-b-PGA). Drug substances with functional groups capable of chelating ferrous ions can be incorporated into CCMs. Please refer line 83-86.

  • What was the safe dose of CCM-CL in the mouse model, and how did it compare to the safe dose achieved with free colistin?

Reply: Thank you for your question. This is addressed in the No-observed-adverse-effect level (NOAEL) section of Result. The safe doses of Colistin and CCM-CL are 8 mg/kg and 13 mg/kg, respectively, when administered by bolus. As seen in the safety trials, using CCM-CL increased the NOAEL dose by 62.5% with bolus injection and by 100% with the slow-infusion mode. Please refer line 269-274.

  • What was the elimination half-life of CCM-CL compared to free colistin in the study?

Reply: Thank you for your question. This is stated in the Pharmacokinetic study section of Result. The half-lives of Colistin and CCM-CL are 12.46 and 102.23 minutes, respectively. Please refer line 282-283.

  • What was the 14-day survival rate of mice treated with CCM-CL compared with that of mice treated with free colistin in the neutropenic mouse model of carbapenem-resistant Pseudomonas aeruginosa pneumonia?

Reply: This is written in the Survival study in the murine pneumonia model section of Result. The 14-day survival rates for the mice treated with CCM-CL, free colistin, and water only were 80%, 30%, and 10%, respectively. Please refer line 297-298.

  • Based on the results of the study, how might CCM-CL become a drug of choice for MDR-GNB?

Reply: Thank you for your question. We conclude that CCM-CL would be a safe and effective choice for patients with MDR-GNB infection based on the good safety profile and its effectiveness against MDR-GNB infection. Please refer line 369-374.

What are the potential implications of the study results for the treatment of MDR-GNB infections in humans?

Reply: We conclude that CCM-CL would be a safe and effective choice for patients with MDR-GNB infection. Further clinical trials are warranted. Please refer line 372-374.

  • What further research might be needed to confirm the safety and effectiveness of CCM-CL in humans?

Reply: Thank you for your question. CCM-CL needs to be confirmed its safety and effectiveness for patients with MDR-GNB sepsis in future clinical trials. Please refer line 372-374.

  • How small and uniform were the dimensions of the individual micelles after the DLS technique?

Reply: The results of DLS measurements indicated that CCM-CL had an average size of 27 nm with a narrow size distribution, as evidenced by a polydispersity index (PdI) of 0.2. Please refer line 237-238.

  • What is the formula for calculating the encapsulation efficiency (EE %)?

Reply: As shown on page 6, line 240-244, an encapsulation efficiency (EE%) was calculated using the following formula:

Encapsulation efficiency (EE%) = (Wi-Wf)/Wi × 100%

Wi represents the total quantity of drug added initially during preparation. Wf is the amount of free drug determined by HPLC.

  • How does CCM-CL compare to free colistin in terms of bacterial load and killing efficacy against CRPA?

Reply: Thank you for your question. The microbiological results were shown in Figure 5. Pulmonary bacterial burden of CCM-CL group was 0 CFU and colistin was 60 CFU at the same dose. Please refer line 309-311.

  • Why is a lower bacterial load of CRPA significant in the treatment group compared to the colistin group?

Reply: Due to slow release of colistin, CCM-CL can continuously inhibit the growth of Pseudomonas aeruginosa, which is partly confirmed by the blood concentration of the drug in this study. This also explains why the survival rate of the CCM-CL group in the animal study was better than that of Colistin. We added this explanation in the Discussion. Please refer line 332-336.

Reviewer 4 Report

What is the significance of infection with multidrug-resistant Gram-negative bacteria (MDR-GNB) on a global scale?

What is the mechanism of action of colistin against MDR-GNB?

To what extent does the toxicity of colistin limit its clinical use?

What are chelating complex micelles (CCMs), and how can they be used in the administration of colistin?

What was the safe dose of CCM-CL in the mouse model, and how did it compare to the safe dose achieved with free colistin?

What was the elimination half-life of CCM-CL compared to free colistin in the study?

What was the 14-day survival rate of mice treated with CCM-CL compared with that of mice treated with free colistin in the neutropenic mouse model of carbapenem-resistant Pseudomonas aeruginosa pneumonia?

Based on the results of the study, how might CCM-CL become a drug of choice for MDR-GNB?

What are the potential implications of the study results for the treatment of MDR-GNB infections in humans?

What further research might be needed to confirm the safety and effectiveness of CCM-CL in humans?

How small and uniform were the dimensions of the individual micelles after the DLS technique?

What is the formula for calculating the encapsulation efficiency (EE %)?

How does CCM-CL compare to free colistin in terms of bacterial load and killing efficacy against CRPA?

Why is a lower bacterial load of CRPA significant in the treatment group compared to the colistin group?

Author Response

We have answered and revised the manuscript according to the questions proposed by the 5 reviewers. In the reply letter, we listed and answered the questions one by one, and the reviewers could refer the revised MS in which revisions were marked by the underlines. The submitted manuscript was edited and proved by English editing company LetPub®. We would appreciate it very much if you and the reviewers could review the manuscript to be published in your journal.

Reply to reviewer 4

 Wei-Chuan Liao et al has reported that the delivery of antibacterial peptide colistin using complex micelle. They have shown the pharmacokinetics behavior of colistin as well as colistin loaded micelle. Finally, they have shown the effect of colistin and colistin loaded micelle in the mice model. It’s excellent work from the group. However, there are several characterization techniques and control experiment are missing. It will be great if author can address my concerns before publication:

  • How that micelle has been characterized? I have seen only DLS data in the paper. I would like to have TEM, SEM, EDAX data with DLS and Zeta Potential (only micelle, only colistin with same concentration used for loading, colistin loaded micelle)

Reply: Thanks for your suggestion. We only did DLS analysis in this study. We will conduct the above-mentioned experiments to analyze the micelle structure in the future research.

  • Although in experimental it is written that the EE has been calculated but I did not find the data. The amount of colistin used in the micelle needs to be addressed, amount of loaded colistin need to be addressed.

Reply: Thanks for your suggestion. As described on page 6, line 240-244, a 100% EE was calculated from the results obtained by HPLC, with no free drug observed. For the preparation of CCM-CL, 120 mg of colistin sulfate was used, resulting in a final concentration of colistin of 6 mg/mL, as indicated in Table 2.

  • The bacterial growth inhibition must be performed with a control micelle i.e without colistin loaded. In vitro experiment will be fine for me to see the effect of micelle. As there is iron ion in the micelle thus it will be great to see if there is any effect of iron in the bacterial growth. 

Reply: Thanks for your suggestion. We consider that in vitro experiment will not provide more information than animal study dose. The microbiological results were shown in Figure 5, showing pulmonary bacterial burden of CCM-CL group was 0 CFU and colistin was 60 CFU at the same dose. The result suggests that iron encapsulated in micelle containing ferrous iron does not adversely affect the outcome in terms of survival and bacterial burden in the infected lungs. Please refer the study results in Fig 4 & Fig 5.    

Reviewer 5 Report

 Wei-Chuan Liao et al has reported that the delivery of antibacterial peptide colistin using complex micelle. They have shown the pharmacokinetics behavior of colistin as well as colistin loaded micelle. Finally, they have shown the effect of colistin and colistin loaded micelle in the mice model. It’s excellent work from the group. However, there are several characterization techniques and control experiment are missing. It will be great if author can address my concerns before publication:

1.      How that micelle has been characterized? I have seen only DLS data in the paper. I would like to have TEM, SEM, EDAX data with DLS and Zeta Potential (only micelle, only colistin with same concentration used for loading, colistin loaded micelle)

2.      Although in experimental it is written that the EE has been calculated but I did not find the data. The amount of colistin used in the micelle needs to be addressed, amount of loaded colistin need to be addressed.

3.      The bacterial growth inhibition must be performed with a control micelle i.e without colistin loaded. In vitro experiment will be fine for me to see the effect of micelle. As there is iron ion in the micelle thus it will be great to see if there is any effect of iron in the bacterial growth.  

Author Response

We have answered and revised the manuscript according to the questions proposed by the 5 reviewers. In the reply letter, we listed and answered the questions one by one, and the reviewers could refer the revised MS in which revisions were marked by the underlines. The submitted manuscript was edited and proved by English editing company LetPub®. We would appreciate it very much if you and the reviewers could review the manuscript to be published in your journal.

Reply to reviewer 5

The authors tried to examine the antimicrobial effects of colistin encapsulated in chelating 2 complex micelles for the treatment of multi-drug-resistant 3 Gram-negative bacteria as well as its pharmacokinetic profile and to compare those findings with free colistin.

  • Abstract should have clearly defined aim of the study. It is obvious comparison, but is should be stated.

Reply: Thanks for your recommendation. Abstract was revised accordingly. The aim of the research was addressed.

  • Introduction: How MDR bacteria is defined?

Reply: MDR bacteria is defined as organisms that are resistant multiple classes of extended-spectrum antimicrobial agents. The definition and its reference are addressed in the introduction. Please refer line 51~55.

  • Aim should be more precisely defined as well as in abstract. Sentences should be more scientific so to speak. It is known what you have done, but it is said complicatedly.

Reply: Thanks for your recommendation. Abstract was revised accordingly. The aim of the research was addressed. Please refer line 27-29.

“We aimed to test the efficacy of colistin-loaded micelles (CCM-CL) against drug-resistant Pseudomonas aeruginosa and compared its safety with free colistin in vitro and in vivo.”  

  • Have you developed your own method? We need some validation parameters, such as accuracy, precision, LOQ, ets.

Reply: We had conducted partial validation for analytical method, and the results were included in the article. The details are as follows.

In the Materials and Methods (Line 143-158)

Analytical method validation for pharmacokinetic study

Sample pre-treatment Five microliters of polymyxin B (Toronto Research Chemicals, CAS number 1405-20-5) dissolved in water (420 μg/mL, an internal standard) was added to 100 μL rat plasma sample. After that 400 μLof acetonitrile and 10% trichloroacetic acid mixed solution (50/50, v/v) was added. The samples were vortex-mixed for 1 minute and then centrifuged at 12,000 rpm for 10 minutes. Following centrifugation, the supernatant was collected and filtered with 0.45 μm PVDF membrane filter for analysis.

Linearity The standard curve was constructed using seven non-zero plasma standards covering the concentrations expected in the study. The original concentrations were then plotted against the responses to obtain the slope, intercept and correlation coefficient (r2) by the least-square linear regression methods.

Accuracy and precision Accuracy was measured as the percent of deviation from the nominal concentration whereas, the precision was determined as the relative standard deviation from the mean (RSD, %). The accuracy results shouldn’t less than 85%. The precision results shouldn’t deviate by more than 15% (RSD, % ≤ 15%) except at the lower limit of quantification (LLOQ) where the values should not deviate by more than 20%.

In the Results (Line 247-260)

Validation of drug level

The equations for the calibration curves of colistin A sulfate in rat plasma were y = 2262273.079x - 310048.238; the corresponding equations for colistin B sulfate were y = 2248789.626x - 160682.478. The linearity of the assay was achieved over the range of 0.38 – 37.91 μg/mL and 0.62 – 62.09 μg/mL for colistin A and B sulfates, respectively, with coefficients of correlation greater than 0.995.

After back-calculating the concentrations of colistin A and B sulfates in all calibration standards from the derived calibration curve, accuracy (recovery) were 90.9% and 122.9%, respectively.

Intra-day precision (RSD%) at concentrations (0.38 μg/ml) of colistin A sulfate ranged between 7.5% and 17.2% whereas, the inter-day precision was 12.6%. Intra-day precision (RSD%) at concentrations (0.62 μg/mL) of colistin B sulfate ranged between 10.7% and 14.0% whereas, the inter-day precision was 12.3%. The results demonstrated acceptable bioanalytical assay accuracy and precision parameters.

  • Do you have ethical permission to condcut this type of study on animal models?

Reply: Thank you for your question. This is written in the Animal section of Materials and Methods. All procedures were approved by the National Cheng Kung University College of Medicine, Chung Hwa University of Medical Technology Animal Care and Use Committee in accordance with the National Institute of Health Guide for the Care and Use of Laboratory Animals and the Animal Welfare Act. The experiments for mice and rats were approved by National Cheng Kung University College of Medicine (IACUC #109307) and Chuang Hua University of Medical Technology (IACUC # A111-03) respectively. Please refer line 184-186.

  • Pharmacokinetic study. Ref 22 should be checked, doi number could not be connected with this authors. Also, I think that text is not referred to this ref.

Reply: This reference was incorrectly inserted and deleted in the revised MS.

  • Results: explain Cmax and Tmax in intravenous models. Also, how do you explain two-compartment model for CCM-CL

Reply: Thank you for your question. Tmax is the time required to reach maximum drug concentration in the plasma after administration of drug. The Tmax is estimated to be 15 minutes when infusion is completed. Cmax is the maximum (peak) plasma drug level after administration of the drug.

CCM-CL is a slow-release drug that acts only after colistin leaves the CCM

carrier, the pharmacological effect is like that of colistin methane sulfonate (CMS),

the prodrug of colistin. Therefore, we use a two-compartments model for analysis

of CCM and one compartment model for colistin respectively based on the

reference 24 and 25. We explain the reasons in the MS (line 202-208).

Round 2

Reviewer 3 Report

I have no further comments

Reviewer 5 Report

The authors have provided all the answers of queries throughly. Now this paper can be accepted in this formate. Congratulations to all author.